# From the Philosophy of Cult to the Philosophy of History in the Work of Pavel Alexandrovich Florensky (* 1882 + 1937)

**Daniel Porubec**

Greek Catholic Theological Faculty, University of Presov in Presov, Ul. Biskupa Gojdica 2, 08001 Presov, Slovakia; daniel.porubec@unipo.sk

**Abstract:** P. A. Florensky dedicated nine writings of his rich interdisciplinary work to the phenomenon of cult, which were first published in a censored form in 1977. We turn our attention to one of these writings called *Cult, Religion and Culture*, published under the common title *Philosophy of Cult*, in which the author elaborates a distinctive concept of the cult as the primary activity of man and at the same time as the gift offered to him for his own sanctification. It is the sacred cult—*sacra* from where, according to the author, two other human activities originate: namely, the ability to create tools—*instrumenta*—and the ability to create abstract concepts—*notiones*. However, both human activities have to be understood as a process of disintegration of the cult—*sacra*. Thus, by prioritizing one of the three human activities mentioned above, we can recognize three historical periods in history. According to Florensky, the human ability to create tools corresponds to the era of *historical materialism*, the ability to create concepts corresponds to the era of *ideologism*, and ultimately, the primary human activity—the life of man in the cult and its culture corresponds to the *sacral materialism* or *concrete idealism*.

**Keywords:** P. A. Florensky; philosophy of cult; philosophy of history; religion; culture

## 1. Introduction

At the beginning of this article, I would like to point to some historical circumstances and key events in Florensky's life which I have already described and published in the journal *Studia Theologica* (Porubec 2019, pp. 197–99). During the sweeping 1917 revolution in Russia, Pavel Alexandrovich Florensky was one of the few Russian religious thinkers who remained in his homeland. After the Bolsheviks closed the Theological Academies, he worked as a technician for Glavelektra (the main administration of the electrical industry), where he studied electric fields and dielectrics (Lossky 1994, p. 190; Porubec 2019, p. 197). From 1920 to 1927, he taught at the State Technical and Art Institute. He was appointed editor of the *Technical Encyclopaedia*, in which he published a number of his articles (Piovesana 1992, p. 337; Porubec 2019, p. 197). This ingenious man, also called the Russian Leonardo da Vinci, a polyglot who mastered not only many living and classical languages, but also the languages of the Caucasus, Iran, and India, was also a geologist, an originator of quarrying in the USSR, a discoverer of a nonsolidifying machine oil later called 'dekanit', whom suddenly the state did not need, sending him into an exile in 1933 (Lossky 1994, p. 191; Valentini 1997, pp. 29–53; Porubec 2019, pp. 197–98). The presence of a priest-scientist in a cassock, which he did not give up wearing even in scientific circles, which also "badly affected" young students, most likely resulted in the Soviets' decision to send him to prison for ten years like a dangerous criminal (Piovesana 1992, p. 337; Hrehová 2001, p. 87; Porubec 2016, p. 381; Porubec 2019, p. 198). Florensky remained in contact with his wife and five children through letters. Visiting convicts was more than a rare occurrence. In 1937, he was deprived of the right to written communication. The last letter delivered to his wife is from June 19 of that year (Florensky 1998, p. 717; Porubec 2019, p. 198). The family did not know anything about him for a long time. In 1958, during Khrushchev's reign, his wife applied for his rehabilitation. She received a message that he

had died in 1943, after 'serving' a ten year sentence. However, it was later discovered that he had died in 1937 (Meň 2005, p. 167; Tagliagambe 2006, p. 11; Porubec 2019, p. 198).

From the secret files of the KGB, we learn about the shameful accusation against Florensky. At first, Father Pavel vehemently opposed the false allegation against him, but after realizing that his confession would release some of his fellow prisoners from prison, he accepted it. He freely chose to sacrifice his own life to free others (Šentalinsky 1994, pp. 171–206; Porubec 2019, pp. 198–99). Florensky's inhuman suffering and ultimately the violent martyrdom could be considered "as a hermeneutic criterion for understanding his entire theoretical and existential concept; from seeking the basis of the truth to destroying at the gulag" (Valentini 2004, p. 14; Porubec 2019, p. 199)

In other words, more than eighty years after his violent death, Florensky gives us a clear example that the witness of Christ is not only his observer or herald, but above all the one who, in his life, recounts and relives the life of Jesus Christ (Valentini 2000, p. 36; Porubec 2019, p. 199). The martyr's testimony is the most precious pearl that man can bring to the treasury of humankind's culture and thus increasingly elevates the world to heaven, to a higher degree of perfection than that which is given by the purely natural way of his existence. In his rich interdisciplinary work, Florensky dedicated nine writings to the phenomenon of cult, first published in a censored form in 1977 (Florensky 1977, pp. 87–248). In this study, I focus on the work *Cult, Religion and Culture*, published under the common title *Philosophy of Cult*, in which the author develops a peculiar concept of cult as a primary activity of man and at the same time as a gift offered to him for his own sanctification (Florensky 2004, pp. 51–78). Within the philosophy of the cult, Florensky thus also interestingly elaborates the philosophy of history, which has always been marked to a large extent by the philosophy of idealism.

At the beginning (in Section 2 of this work) of his reflections, Florensky points to the cult as man's primary activity, and he proves his argument by analyzing man's two natural activities. They are the creation of tools in the technical sense, that is, things with a certain purpose or meaning, and the creation of concepts, intangible tools, such as words. However, as the author shows later, both have a certain insufficiency in defending their objectivity. Therefore, he searches further and concludes that there must be another, higher activity in which the human combines with the divine, and that is the cult.

In the following section, entitled *From God's Work to Homo Liturgus*, the author opens the argument of the divine work, *theurgy*, as the only source not only for the cult but also for the resulting human activities of creating tools and concepts. He draws attention to the need for ontological connection of man to the source, aiming at the unity expressed in human integrity. Florensky already clearly understands that man as a historical, creative being seeking self-realization receives and transforms the divine on the natural level and creates a concrete culture and society always in a certain time frame and works with concrete tools, concepts, and the cult. In other words, these three activities take on concrete forms and shape the historical process.

In the next section, I show how originally and ingeniously Florensky outlines the three images of historical processes corresponding to the three abovementioned activities of man, according to which of the three is given priority. At the same time, Florensky warns against the danger of counterfeiting or fictitious activities and subsequent deviations in a society, which come either from not finding the right balance of these three activities or from their deliberate manipulation. According to Florensky, this could have, and indeed has, catastrophic consequences in the social sphere including the individual and personal reality.

In the final section, I present how Florensky recognized and reacted to the contemporary and purely intellectual knowledge of cult and religion at the beginning of the 20th century. The author himself chose two representatives of the modern views, É. Durkheim and W. R. Smith, and explains in what way the two misunderstood the subject, but at the same time, how they correctly confirmed the importance of the cult for human history. Overall, he distances himself from the views of both.

Marginalia on the style and language: Florensky's style and terminology of are not easy for everyone to read. He himself is often carried away by the sound of words and also brings lyric and rhythm to his definitions and arguments. Translation from Russian into other languages thus loses this feature, as it is necessary to keep the argument clear and not to strictly adhere to external expressions and sounds. Nevertheless, I also tried to find ways to present his linguistically colored work and its contribution. In addition, the author often seems to revolve around the subject, clarifying it from every angle, which sometimes seems tedious or irrelevant to the reader. Florensky, however, does not write impulsively but with the premeditated logic and an approach of a mathematician. In his reasoning, he always keeps the goal in mind and makes it clear either at the beginning of his argument or proceeds to it chronologically with the eye and the mind of a reader (Žák 1998, pp. 56–65; Žák 2003, pp. 598–614; Žák 2018, pp. 415–32; Attard 2020, pp. 219–22).

## 2. The Cult—The Primary Human Activity

There is no human being in the world, says Florensky, who would not be Platonic for even a moment and did not ask about the hidden essence of things or did not look for a kind of mystery behind the observed object. The author asks: "Has there been anyone who, with the help of *eros* has not fathomed to the human judgment inaccessible depths of knowledge? Who has never experienced the fall of the impassable walls between the object and the subject—when the 'I' goes beyond its egoistic limits . . . and becomes one with the whole world?" (Florensky 2000, vol. 3(2), pp. 145–46). Father Alexander Men notes that Florensky liked Plato, whose philosophy was fully in line with the author's childhood vision of the world. In the vision, the invisible is the source of the visible world constantly communicating with us (Attard 2020, pp. 214–18; Porubec 2018, pp. 221–22). "All his life Pavel Florensky loved Plato, studied Plato, and interpreted him. And it should be added that there is nothing special about it. The English philosopher Whitehead expressed the opinion that global philosophy is actually only a commentary on Plato. Platonic thinking once and for all affected the main directions of the human spirit and human thinking." (Meň 2005, p. 171) All philosophical currents are as if of blood relation with Platonism. Idealism is the basis of every philosophy, through which both European philosophy and culture breathe. Where did Plato find the concept of *eros*, as the beginning of every philosophy, as the key to all the mysteries of the world and to the kingdom of truly existing, immutable, immaterial images of being? Florensky is convinced that the answer to this question cannot be found in Plato's philosophical training or in any abstract principles of thinking, but in the fertile soil of the "national soul", that is, in the universal consciousness, i.e., something what is inherent to all (Florensky 2000, vol. 3(2), p. 147). However, what is inherent to all people without distinction is above all the one activity, more precisely the life in a certain cult and with the cult.

Among all the human activities that appear to be a kind of cultural activity, Florensky notices, exactly for the reason mentioned above, the cult. Similar to any human activity, the cult is carried out thanks to the tools of the cult, which include the temple and all the other elements of the cult, such as chants, prayers, etc. Man himself is ζῶον τεχνικόν, the living being creating instruments. It is in this specification where human uniqueness in comparison with other beings lies. Plato is the one to whom the use of the word ζῶον in connection with God's being is attributed. "Θεὸς ζῶον ἀθάνατον"—God is an immortal living being (Platone 2005). Only God, angel, and man can be spoken of as a living being—ζῶον (Lat. *animal*, the opposite to this word is the word θήρ—an animal, Lat. *bestia*). It is also worth mentioning Florensky's remark that an instrument (Greek Ὄργανον—organ) is an external projection of the inner creative forces of man (Florensky 2000, vol. 3(1), pp. 378–79), the forces which form all his empirical being, that is his physical and mental life (Lanfranco 2019, pp. 23–34; Boneckaja 2010, pp. 90–109), while these mental–bodily instruments of being "are instruments of the spirit, created by the spirit for itself, at the same time ours, created by us . . . " (Florensky 2004, p. 52).

By the word instrument, one means above all the material instrument of technical culture, e.g., plough, hammer, saw, etc. We call this type of machines, tools—*instrumenta*. It is true that in addition to this type of tools we also have tools that literally are the least material and 'airy', but no less effective, and they are *words*, as technically formed concepts and terms. "The word, 'the airy nothing', is also an instrument of the mind, without which the mind does not develop and is not realized. Not in a figurative sense, but in the true sense of the word, words are tools." (Florensky 2004, p. 52). The author simply calls this type of tool concepts—*notiones.*

Florensky then immediately asks whether perhaps man, as a living being creating tools, is fully realized only in the mentioned range of creating tools—*instrumenta* and intellectual concepts—*notiones*, or is there another activity, more precisely the area of his interest, beyond the presented instrumental—conceptual boundaries? It is indisputable that the instruments—*instrumenta* stand before us as a fact that cannot be doubted, they simply exist. However, their wisdom or meaning—λόγος must be proved because it is not given immediately. In other words, a tool is simply a tool only if it is sensible, otherwise there cannot be a question of a specific tool. On the other hand, the notions—*notiones* are accepted by us with full conviction, as sensible facts—λόγος, but, their feasibility and their ability to incarnate in a visible (instrumental) form must be proved because it is not given to us immediately. "Machines—instruments appear as ἔργα, as things, but their sensible ἐνέργεια that has created them must be proven. We are aware of concepts—terms as sensible ἐνέργειαι, as facts, but whether they are also ἔργα is not seen without a proof." (Florensky 2004, p. 53). The Greek word ἔργον has multiple meanings. In the examined context ἔργον here represents a thing, object, or object of knowledge. The word ἐνέργεια means actuality or action (Romizi 2001, pp. 441, 532; Ferrari-Bravo 2016, p. 33; Ferrari-Bravo 2000, pp. 118–26; Lingua 2003, p. 14; Lingua 1999, p. 26). Florensky takes the terms ἔργον and ἐνέργεια from Aristotle, who already says in the well-known Nicomachean Ethics: "every action and pursuit, is thought to aim at some good; and for this reason the good has rightly been declared to be that at which all things aim. But a certain difference is found among ends; some are *activities*, others are *products* apart from the activities that produce them"[1] (Aristotle 1994–2009). Aristotle himself gives us an explanation of these concepts as follows: "For the activity is the end, and the actuality is the activity; hence the term 'actuality' is derived from 'activity', and tends to have the meaning of 'complete reality'"[2] (Aristotle 1924). According to Florensky, the creative power of reason thus divides into the production of things whose meaning is invisible and the production of senses—meanings as purely rational facts, the reality of which again cannot be recognized immediately. It would be possible to prove the meaning of things and the objectivity of meanings standing on the basis of a thing with an immediately given meaning, or a meaning, the reality of which would no longer require a series of evidence.

According to Florensky, it is necessary to have even only one cultural human activity, where the visible unity of its two poles is given, that is, the indisputable incarnation of meaning or the indisputable spiritualization (одухотворенность) of things. If there is no such unity, then the unity of one's own consciousness is not possible, i.e., transcendental apperception, or, the self-identity of the I (I = I) and its own reason would disintegrate (the I would tragically split from itself), which leads to an ontological madness or hell. In other words, the unity of self-consciousness necessarily presupposes such an activity of reason (both practical and theoretical) which is a harmonious unity between the reality of *instruments–things* and the rationality of their *concepts–terms*. The author unfolds the next argument as follows: "Therefore, the condition for reason itself to exist is that it must rely on a living antinomy and maintain its balance within the antinomy. ... The antinomic activity of reason is not demanded only by itself, but it is also necessarily an indispensable condition of life as such, life in its full dimension." (Florensky 2004, p. 55). Florensky also presented a closer explanation on the history of the origin of the term antinomy used by philosophers such as Sophocles, St. Augustine, finishing with Immanuel Kant in his most famous work, *The Pillar and the Ground of the Truth* in the article *The Historical Development*

*of the Term "Antinomy".* In short, we could say that the word antinomy has rather juristic than philosophical meaning. It expresses the internal self-contradiction of a law. It was only thanks to Immanuel Kant that it became established in the field of philosophy. For Pavel Florensky, the term antinomy takes on different semantic dimensions. The word antinomy becomes synonymous to the word dialectics. In the process of true knowledge, the interplay, or rather the synergy of the opposites, is necessary. Florensky describes it as the "art of dialectics" and at the same time as the art of the Christian life, which, however, remains a mystery to man himself for one reason only. Quoting St. Macarius the Great, the author says: "The Truth itself (God) encourages man to seek the truth." The word antinomy used by Florensky is then diametrically different from the antinomies of pure reason as Kant tried to define them in philosophy. Of course, pure reason cannot refer to anything higher than itself (Florensky 1990, pp. 489, 582–84). The importance of this balancing between the two poles is confirmed by other authors, such as Kierkegaard who recognizes its crucial role: "A valuable lesson for us (for man) rests in learning to live with a creative tension between immanence and transcendence." (Valco 2016, p. 98)

That said, where should we look for this living antinomy, this primary activity of human existence, as the author says the activity *prius*, through which man becomes man? This cultural activity must, of course, have its material basis in the world, but at the same time transcend it, i.e., it is at the same time a spiritual activity, similarly as concepts and terms appear as '*nihil audibile*' but in fact exist immediately in our minds. At this point, Florensky provides clarification to the conclusions he has formulated. The given balance between the real and ideal pole of human activity is the cult because it is the incarnate meaning where the word became the body "ὁ λόγος σάρξ ἐγένετο" (Jn 1:14). Thus, it is no longer an exclusively human activity, but God–human synergy, where man constantly finds the balance for not only his knowledge and being, but the Knowledge and the Being, in better words, find the balance thanks to the God–Man Jesus Christ. The cult is thus a place of unification of the upper and lower, Heaven and Earth, a symbol par excellence, said in Plato's language, a revelation of an idea, an embodied idea (Florensky 2004, p. 55). In addition, Florensky quotes here a few examples from the liturgical texts of the Byzantine Church of the Eastern Rite, where the cult is identified with the Mother of God. 'Today, the birth of Christ united the earth and the heavens. Today, God has come down to earth to bring man into heaven.' On the Feast of the Annunciation, we sing: 'Today is the Feast of the Annunciation ... Earth and heaven have united.' (*Večiereň na nedele a sviatky* 2010, pp. 265, 305). The place of connection between heaven and earth is the Mother of God. The Mother of God is a noumenal, spiritual ladder, by which God descends to earth and man ascends to heaven, as the author says: "The Mother of God is the Church, the centre of the Church, the ecclesial heart, in whom particularly clearly the activity which forms the deepest basis of the own consciousness takes place. This activity is the ladder by which God descends into the world and man ascends to heaven, by this activity the lower and the upper always unite. Therefore, its result is the illuminated matter, transformed matter. In the process, the reality is spiritualized, sanctified, deified. In the Mother of God, in this new reality, as in the central node of being, both practical and theoretical activity find their place, while it becomes obvious, that it is no longer two activities, but one activity of real meaning or meaningful reality. It is one and the same activity, but in its own interconnected momentums." (Florensky 2004, p. 56).

## 3. From God's Work to *Homo Liturgus*

The existence of the cult, that is the third of human activities analyzed by Florensky and called as primary, is also witnessed by art, e.g., painting. Looking at the artistic painting no one doubts about its materiality (perceptibility) or about its ideal meaning. The work of art is immediately materially given as ἔργον, together with its spiritual power, ἐνέργεια. Art is a true testimony that there is a third reality we are questing after, i.e., the cult (Florensky 2004, p. 56).

Florensky points out that art fell out of the nest of the greatest art of all arts, that is, God's work: theurgy. Theurgy, according to the author, "was the maternal womb of all sciences and arts. She was the real condition for the development of human consciousness, the mother of human life and man's authentic activity." (Florensky 2004, p. 57). When man ceased to recognize his dependence on God's action, the unity of human activity gradually disintegrated and theurgy narrowed only to ceremonies, i.e., into 'a cult' in the later sense of the word. All other activities of life, breaking off from God's action, have legitimized their fictitious existence as their own independence. The content of these activities ceased to exist as something unconditionally valuable and immutably real, and as a result, the form and content split up and subsequently began to unify only accidentally, tendentiously. Human activities in and of themselves ceased to be comprehensible in the order of life and thus inevitably gravitated to the order of death; more precisely, they became a momentary taste, fancy, whim, pleasure, simply fiction, not at all coming out of necessity, but performed intentionally, on purpose. From Reality and Meaning (theurgy), we have moved to realities and meanings separated from the First, and as a result, these realities have become empty and the meanings deceptive.

"Things have only become utilitarian (useful) and concepts have become only convincing. However, the utility is not a sign of reality and the urgency is not a sign of truth. Everything has become just similar to the Truth, ceasing to be a companion of the Truth, ceasing to be the Truth and in the Truth. In short, everything has become mundane (secular). This is how the Western European human civilization arose—the rot, decay and almost the death of human culture. In this way the culture disintegrated and decomposed into particularities, details and individualities, none of which is necessary, but all random and unstable. The human person similarly disintegrated, losing the necessary condition of one's own unity and along with it lost the power of noumenal consciousness (self-consciousness). In this way, the soul disintegrated into the summa of thoughts and perceptions (or rather impressions), i.e., the conditions blown in by random winds from outside." (Florensky 2004, p. 57).

This seems like a harsh critique of Western culture and philosophy, while the tendency of Russian philosophy is still toward the unification and unity. "Western philosophy's struggle for the rational world view was continuous and painful. Metaphysical knowledge has evolved on the path of 'abstract principles' for centuries. Russian philosophy considers the triumph of 'integral knowledge' as the most important epistemological task. Such knowledge must embrace the world in all its diversity and indivisible fullness, mobilize all levels of human consciousness and synthesize all methods of world comprehension." (Kortunov 2014, p. 47). The fragmentation is apparent not only in Western European culture, but in culture as such, as a consequence of its detachment from God's action. Florensky also blames those philosophical systems more or less rejecting the real numerical identity and unification of the two worlds visible and invisible and thus denying the cult. The absence of the possibility of the cult as a real unity, on the other hand, creates only imitations of the cult in every direction of human activity. The author therefore divides all philosophical systems into two groups. They either recognize the consubstantial unity (όμοουσία) or only similarity (όμοιουσία). Only the philosophy recognizing the unity is the philosophy of the true culture, i.e., "of the idea and true reason, the philosophy of the person and true self-realization." A philosophy recognizing only the phenomenal similarity and not the numerical identity is "the philosophy of matter and rigid stagnation." (Florensky 1990, p. 80).

The philosophy of *oneness-in-substance*, also called unitotality, which had been present since ancient times in the work of philosophers such as Plato, Aristotle, and Plotinos, was also developed by many church fathers. In the modern philosophy, it is included in the concepts by Schelling, Fichte, and Hegel, and in Russian philosophy, especially in the work of Vladimir Sergeyevich Solovyov (Lossky 1994, p. 203). Father Serafion Maškin also had a great influence on the author with regard to the mentioned concept of unitotality, although we can learn about his writings only from the articles by Florensky

(Florensky 1990, pp. 619, 791). Florensky's merit lies primarily in the conscious use of the term *oneness-in-substance (ὁμοουσία)* in both metaphysics and cosmology. In other words, Florensky clothed his philosophical and cosmological visions in a religious garment of theological terminology and thus has nominally solved the problem of 'unitotality', 'universally-human consciousness', or 'universally-human principles' so close to the hearts of many Slavophiles (Zeňkovsky 2001, p. 837).

Thanks to the cult, in which there is a real, numerical unity of the two worlds, it is possible to talk about the self-realization of a person. Florensky speaks of man as *homo liturgus*, because man himself, just like the cult, is an antinomy, i.e., the living unity of the infinity and finiteness, eternity and timeliness, unconditionality and transience, necessity and possibility, the knot of the world tying the ideal and the real, and thus cannot act otherwise than according to himself, in his likeness, that is, creating such contradictions as he is himself. All other activities, apart from the cult activity, express man's character only one-sidedly; with far less clarity, that is, they incline within the abovementioned antinomy either to one side or the other and thus create the already discussed things without meaning or elaborate concepts—meanings without reality (Florensky 2004, pp. 58–59).

If man were to produce things without meaning or developed meanings that would not find real expression, one would thus lay the fictitious basis of a nonhuman, noncultural activity, an activity of a nonspiritual being, the activity that is already beyond man's scope. To unite God and the world, spirit and body, and meaning and reality is not an exclusive mission for religion, more precisely the cult, but a task for man himself. Therefore, the cult, as the liturgical activity of man, in a logical not in chronological order, is the activity *prius*, because the incomprehensible and mysterious unification of the two worlds occurs in it. Florensky thus explains the three cultural activities of man: theoretical activity, which creates concepts-terms (**Notiones**), practical activity, the product of which are machines and instruments (**Instrumenta**) and liturgical activity of man, activity *prius*, which is finalized in the cult in the Sanctuary (**Sacra**). "Notiones, Instrumenta and Sacra are the three types of creative human activity." (Florensky 2004, p. 60).

If we go back to the primary liturgical activity of man, as we have already mentioned, to the activity *prius* in the logical order, i.e., to the cult Sacra-S, we find that it is a concentrated and focused unification of two activities, namely Notiones—N and Instrumenta—I. If activity S is an activity *prius*, it necessarily follows that activities N and I must be from the aspect of the general evaluation of the historical process understood as moments of division or disintegration of Sacra—S. Florensky proceeds to specify these activities in more detail, placing them in a historical context, which undoubtedly makes his work original. The author asks: "What is the real-historical principle of their mutual relationship? What is the genetic relationship among S, I, and N?" (Florensky 2004, p. 61)

Subsequently, Florensky calls the S, I, and N theories by their more relevant names, projecting them into the historical process. Therefore, in the horizon of history the liturgical, practical and theoretical activities of man correspond to the historical theories of the Cult, Economy, and Worldview. The triad of human activities suggest the possibility of three basic theories of the historical process, depending on which of the three activities is considered primary.

## 4. Theories of the Historical Process

### 4.1. The Worldview—Notiones

When we prioritize the activity of reason, in other words, the Worldview, we mean the domination of a certain reasoning system or rather a set of concepts about the world, morals, law, opinions about God—in the form of various mythologies, even doctrines or dogmatics. Concepts and terms are tools of the Worldview that serve to create further concepts and judgments. Such an activity of reason, of course, always turns only toward itself. Precisely in there, it finds the only source of its growth and its own exclusive incarnations and realities. The result of prioritizing reason is the appearance of 'ideologisms'. An ideology looking at the other two human activities—the economy and the cult—sees merely a kind

of an application in the forms of economy and in the rites of a cult and use of scientific, mythological, and dogmatic capacities of the reason itself. First of all, there is an idea or a worldview, and only later, the tools are constructed or the ceremonies for the cult invented. In other words, everything is just a mirror reflection of a certain thought process. However, the author detects a great danger which stems from an actively reflected ideologism. He explains that if ideologism succeeds in intentionally changing people's thinking, it also succeeds in creating the corresponding tools and cult all in unison to that thinking. No ideologism is ever satisfied with the fact that reality is a pure reflection of its ideas, but it goes further and creates the reality to rule over and transform the reality into its own chimerical image. The conspicuous feature of an ideology is to emphasize the individual abilities of an individual person and subsequently create a personality cult. The theoretical activity of reason is, of course, individual; therefore, it needs great heroes of the time, politicians, national figures, creators of systems of economic organization, and also religion reformers. According to Florensky, the theory of ideologism, when all areas of life were to be subject to rationally constructed schemes, culminated in the period of rationalism of the 18th century. The enlightenment absolutism was the price paid for the error of ideologism and eventually culminated in cruel terror (Florensky 2004, pp. 62–63; Šoltés 2020, pp. 96–99).

### 4.2. The Economy—Instrumenta

The relationship among the three human activities can be visualized in another way, that is, if the primary activity is not the worldview but the economy—country management. According to the author, the economy is a complex of tools ensuring the functioning of the external material-utilitarian culture. However, we can still divide the mentioned tools into tools in the true sense of the word and into weapons, while the first serve to acquire material wealth, and the second serve to protect them. Everything else is subordinated to the economy as a material-utilitarian sphere, including thinking, which is in the hands of the economy that already creates 'its' tools and 'its' weapons, often deliberately confusing the two, as well as a cult that becomes a kind of tradition or rather an inherited folklore, resulting from a purely aesthetic-material need for the realization of man. As a person decorates one's own dwelling with various objects, he simply needs to embellish and thus materialize the desire for a kind of inherited infinity. However, the 'value' lies solely in the material tradition and not the kind of infinity. We can clearly see that the whole reality is adapted to the image of the instrumental economy, where the matter has some value, and everything else only follows from it, and as it is shown later, everything literally parasitizes on it. According to Florensky, this inevitably gives rise to the need to absolutize, sanctify, or we should say deify, the tools of the economy. While in the previous system with the priority of the worldview, the cult of personality dominated, in the case of the priority of economy, the deification of technical tools occurs, for example, agricultural tools, scythes, ploughs, harrows, or vineyard presses; labor tools such as hammers, screwdrivers, wrenches, and various weapons; and also plants, like an ear of wheat, or domestic animals, and finally a home, the cultivated field, the forms of the public management, or power. It is indisputable that we have reached the platform of the proponents of historical or economic materialism. Spiritual realities are here undesirable, which is well expressed in the following excerpt: "People are invited to celebrate their freedom of choice, ridden of the burdensome task of a true self-reflection. They are to devote their time and energy into solving practical issues 'at hand and shy away from the impractical issues' of spiritual integrity and deep moral responsibilities. These seemingly less tangible realities become less and less intelligible and increasingly perplexing, as individuals lose grip with the inner core of their being (their 'authentic selves'), which urges them even more to flee into the more intelligible 'and real' world of economic choices and instantly available gratifications. Thus the vicious circle of economic realities intertwined with human insatiable desires and unquenchable fears closes in upon us." (Valco 2015, p. 135) Therefore, according to Florensky, it is necessary to mention that the economic process, in accordance with the

teachings of historical materialism, takes place spontaneously, by itself, governed by the fateful laws of economics, which are not known and no longer understandable by human mind. For this reason, it is no longer the individual national leaders who determine the direction for the future but natural, shapeless masses of people. The people demand this and that, the nation wants . . . The people or the nation is the above all standing *deified principle* in general. Under the disguise of technology advances and the natural progress of allegedly unidentifiable masses, the value of a human person and personality is devastated. Already in the middle of the 19th century, precisely thanks to historical materialism, the foundations for the removal of any authority of the emperor, leader, or priest were laid which subsequently historically happened (Florensky 2004, p. 63).

### 4.3. The Cult—Sacra

Finally, we mention the third that is the liturgical activity of man, which we simply named the cult. According to Florensky, the cult—Sacra is a certain summary of sanctuaries, i.e., holy things (instruments) and such activities and words (notiones), including relics, ceremonies, and sacraments, which serve to create our relationship with another world, namely with the spiritual world (Florensky 2004, p. 62). Elsewhere, the author defines a cult as "an extracted part from the whole of the reality where the immanent and the transcendent, the lower and the upper, the present and the future, the timely and the eternal, the conditional and the unconditional, the disintegrating and the immortal meet." (Florensky 2004, p. 30).

To find the fulcrum of our being and our knowledge, to find the very source of life in which death would drown is possible, according to the author, only in the cult concentrated in the truth about the incarnate Lord Jesus Christ, **the Pre-eternal Word—the Logos**. Jesus Christ is the absolute meaning (*notiones*) of our life (**Everything** was created in him) (*Conciliorum Oecumenicorum Decreta* 1991, p. 5); he is the absolute reality (*instrumenta*) (Everything **was created** in him). Every cognitive process is either based on this way, truth, and life or is simply false and unreal. The face of Jesus Christ is the Incarnate Meaning and the true foundation (the orientation) for the mind. Cult is the concrete realization of this foundation. In the cult, our mind finds life, finds the concrete and at the same time real categories of its own existence, and finds the way and the truth. Beyond such cult, as the real representation of Jesus Christ—the Meaning of all meanings (compare Jn 1: 3), there is nothing truly spiritual and reasonable; there are no true words, no ethical principles. The author adds: "The Incarnate Meaning—The face of the Lord Jesus Christ is the true basis of the orientation of thought. The cult represents a specific development of this orientation. The unchangeable elements of the cult are the Christian categories: they are concrete and real at the same time, corresponding to the orientation itself. Such categories are cross, blood, light, etc. (...) The Lord is a Man among many other people, but at the same time He is the Only-Begotten Son of God and the Consubstantial to the Unconditional and Eternal One (God). All the values of the spirit are such that they have the ability to be much more than they are in themselves, and so they are symbolic." (Florensky 2004, pp. 115–16) The cult is "the true fullness of the true life". This fullness "can be thought of and pondered over and over, can be approached again and again and never exhausted." (Florensky 2004, p. 127). The mysteries of the cult, i.e., the Sacraments constantly nourish our mind and give birth to the mind by their (sacramental) meaning (λόγος). "The one truth, in its fullness, I dare say more, the only objective truth offered to man are exclusively the mysteries—the sacraments, which in the strict sense of the word deserve the predicate truthful." (Florensky 2004, p. 131).

That is why Florensky does not hesitate to emphasize that "all life must be bordered by the cult, concentrating around its unconditional Center—Golgotha and the Resurrection. From the cradle to the grave, all the moments, all growth, all the life struggles, all the facts, all the movements, all the efforts, all the words, everything, even the smallest and the slightest motion, should be cult-centric, gravitating toward its center, as every particle is attracted by the center of gravitation in the Solar system." (Florensky 2004, p. 156) We

can further develop Florensky's intuitions. Only in this union with Christ and in Christ within the cult (sacra), which here is the man's response to God's voice, originates, and is achieved the harmonious union between the materiality of the meanings (instrument) and the meaningfulness of things (notiones). Christ is the true way (instrumentum), the truth (notiones), and the life (sacra). This unity of the three human activities can be subtly falsified by his opponents who, in contrast with Christ, by creating their own 'tools', 'concepts', and 'cults' deprive man of his primordial image—Christ who is the only and full realization of a person's freedom. Religious materialism is a very dangerous and cunning, purely intellectual 'substitute' for faith in Jesus Christ who appears in the cult.

### 5. Confrontation with Sacral Theory

At the beginning of the 20th century, a new theory appeared called Sacral Theory, or, concrete idealism. Its opponents call it religious materialism. This theory emphasizes the central position and the necessity of cult for the possibility of existence of a religion itself. The cult ritual does not come from a myth or dogma, or from the external rules of the cult, but on the contrary, all the other activities are only a multiple layering of cult activity. From this theory comes a clear conclusion that the myth itself, the dogmas themselves, or the rules of the organization of the cult themselves lead to the secularization of religion and are *de facto* the moment of its disintegration (Florensky 2004, pp. 64–65) In this context, Florensky observes the statements of William Robertson Smith, one of the proponents of the sacral theory. Smith presented the opinion that cult, which is a practical custom, meant everything in ancient religions and the cult practice preceded its doctrine and theory, just as people create and live the general rules in the society organization before these rules are expressed as general principles in a written form. The same is true about any religious belief, which exists prior to a religious theory (Smith 1901, p. 20).

Florensky clearly distances himself from the sacral theory because this theory is evidently built on a positivist-empirical basis, which is foreign to religion. It is true that it correctly reveals the root of the problem, i.e., the fact that the cult is the central, primary activity of man. However, after the discovery of the cult as the central nerve of religion, sacral theory leaves it at the mercy of a purely rational knowledge lacking life which streams from the cult. This theory is just a certain worldview about the cult and not the life in a cult. According to Florensky, the representatives of sacral theory became lost within their intentions but remained true in what they actually pointed out, i.e., the importance of a cult. The view of an enemy of a religion is often more complex than the view of a neutral and disinterested critic because the enemy sees the core of the problem deeper and more immediately. Undoubtedly, among such opponents of religion is the French philosopher and sociologist Émile Durkheim, who spoke about the necessity of a cult for the believer and expressed it, so to say, 'in golden words': "Anyone who has actually practiced religion certainly knows that it is a cult that evokes feelings of joy, inner peace, clarity, enthusiasm, which together represent for those who believe something as exact proof of one's own faith. A cult is not simply a system of signs through which faith is manifested externally, but a group of means by which faith is created and periodically re-created anew." (Durkheim 1912, p. 596) Durkheim did not grasp the relationship between faith and the cult correctly but could not deny its central role. Such is the importance of a cult from the point of view of the antireligious theory. If a positivist opponent puts such weight on the cult, we can only readily agree with his advocate—Florensky—who, looking at the cult claims rather poetically that our past did not know abstract concepts, because it constantly thought and spoke in the concrete images, it "spoke" the cult.

In other words, the cult was the center of human life, and no interpretation was needed. Similar to poetry, the cult is the fruit of an immediate knowledge. The poet does not use abstract concepts but specific images. No poet ever thought about writing a commentary on his own poem. On the contrary, such a comment would testify to his inability, more precisely to the imperfection of the used means of expression. Any clarifications of poetic images are generally a later work and were not necessary for the author's contemporaries.

In this way, the cult did not need additional comments in the past, because by the process, it would have slowly become a purely human and imperfect act. One would have the opportunity to study the cult as if in a laboratory that is 'from the outside', and would not live in the cult and from the cult. Again, there would be a fraction between the knowledge and being, the theory and practice, the meaning and the specific thing, which would result in the creation of things without meaning and meanings without the proper incarnation, i.e., such "facts" that are already beyond or out of the horizon of our knowledge and being, i.e., they are nonhuman (Florensky 2004, p. 75).

Nevertheless, the question arises as to whether Florensky's cult-centric views are not too exaggerated and suspiciously smell of excessive irrationality. Science values reason. It is the systematic manuals with predetermined deadlines in the field of the given science which are necessary for a contemporary scientist to reach deeper understanding of the facts examined by reason. With this in mind, how, if not by reason, will one seek the truth and even, according to the author, be able to know the truth in an unmediated way? Through what do we know, if not through reason? The author certainly did not want to point out the meaningless effort of intellectual cognitive abilities. After all, Florensky himself is the author of more than forty patents in the field of exact sciences. The broad-spectrum of his intellect, together with all of his inventions, would thus present a great absurdity and so there can be no question of the irrationality. The author would deny his own experience as well as his own reason. If Florensky is not wrong, there is only one possibility left, and that is the one of which the author himself is well aware, i.e., that the truths we seek are constantly revealed to us by the Truth itself. Our desire to know the truth is real only because there is the constant and own desire of the Truth to reveal itself. The mind is thus able to know the Truth but does not have the strength to live in the Truth; the mind does not have the ability to defend itself through self, but it presupposes the grace of the revelation of the Truth; it necessarily presupposes the Savior and life in Him. Without this postulate of the revealing Truth, Florensky will remain misunderstood or misinterpreted. Life in God, life in the Truth, is possible through the cult that is through such space and time, which is the unmediated revelation of the Truth, which is the unity of the visible and the invisible. The approach of unity between knowledge and faith without contradiction is presented also by A. S. Khomyakov: "Faith is always a consequence of a revelation identified as a revelation, it is contemplation of an invisible fact that is revealed in a visible fact; faith is not a conviction or logical convincement based on conclusions, it's something much greater. It's not an act of cognitive ability only which is estranged from other abilities, but an act of all forces of reason which is gripped and profoundly captivated by a living verity of a frank fact. Faith is not something what we conceive or feel, it is something we conceive and feel at the same time; in short, it is not only cognition, but cognition and life." (Khomyakov 1907, p. 61).

The fact that man went on the scientific journey of searching for a purely rational person without a transcendence, which our pure reason does not understand because the transcendence exceeds the reason, resulted in the creation of a 'cult of pure reason' into its own 'transcendence'. Florensky is convinced that the concepts of these purely rational sciences must be spoken of as the products of the disintegration of the true cult, where the ceremony disintegrates into **a meaning—a notion** and **a thing—an instrument**. In parallel with the process of growth in material technology, the purely intellectual knowledge also grows, because both processes are in fact only one and the same process, i.e., the process of the disintegration of religion, the disintegration of the cult. From here, it is understandable why the appearance of science and technology turn out like the primary cause of the disintegration of theurgy. Theurgy is in its all-encompassing meaning narrowed into a rite in the narrow sense of the word, just as if the holistic life were narrowed into a crystalline basis. The ceremony itself loses its significance which, however, before the cult breaks down into a **meaning** and a **matter**, is objectified in **myths**. Myths are understood by Florensky as testimonies about movements or mental processes that are born within a religious consciousness (Florensky 2004, p. 81). The author thus defines in an abbreviated

form what the linguist Potebnja said about myths: "A myth is a verbal expression of such a clarification (apperception), by which the clarifying image, having only a subjective meaning, is attributed objectivity, that is, the real being." (Potebnja 1914, p. 503) Man clings to the external ceremony—the myth, and does not capture the essence; he confuses the partiality for a whole, which in turn causes death to the myth itself. The myth causes that the ceremony is not received as a revelation but indirectly and thus extinguishes the cult to the point that it kills it. However, as a result of that the myth also dies because it exists only due to the life-giving cult (Florensky 2004, p. 76).

The already-mentioned ideologism, which presupposes a unified ratio or historical materialism, which prioritizes the activity of the formless masses, are, in the understanding of Florensky, the product of the disintegration of the cult as human activity *prius*. Thus, the three ways of relationships among the human activities can be schematically expressed as follows:

1. N → (S, I). Ideology
2. I → (N, S). Historical materialism
3. S → (N, I). Sacral materialism or concrete idealism

The parentheses indicate those human activities that are perceived as secondary, and the direction of the arrow indicates where from the activities derive. Florensky cannot deny being a mathematician in this case either, because he logically creates other schemes of possible relationships between human activities, which we can formally assume from the three mentioned formulas. Each of the three activities, which is the starting point for the other two, is divided into three other possible modes of interrelationships between the secondary activities, depending on whether these activities are granted full independence or one of them comes as the first. Subsequently, this creates nine possible points of view on the origin and the mutual relationship among the worldview (N), economy (I), and cult (S) (Florensky 2004, pp. 77–78; Florensky 2000, vol. 3(2), pp. 381–83, 439–41).

## 6. Conclusions

In the remarkable and unique title and content of *Philosophy of Cult*, typed in 1922 and compiled from the author's lectures from 1918, Florensky remains faithful to the philosophical conception of idealism, with the cult as its source (Komorovský 2011, p. 135). This fact needs to be emphasized once again. It is not idealism that is the source of the cult and the cult an expression of man, but on the contrary, the cult is the source of idealism. The cult is the fertile soil for the human soul. It touches every individual soul fulfilling its most hidden desires. God's cult imprints unique forms of thinking into each person in one's own personal history (Porubec 2019, pp. 197–224; Žák 2016, pp. 169–71; Tagliagambe 2016, pp. 44–50).

The early period of Florensky's work was influenced by the spirit of Plato's idealism. Already before the author's most famous work *The Pillar and Ground of the Truth*, also marked by the sacralization of Plato's idealism, the author attracted considerable attention with his pro-Platonic orientated thinking in the article "*The All-human Fundamentals of Idealism*" presented to the auditorium of the Moscow Spiritual Academy in 1908 (Gavrjušin 2005, p. 240; Dancák 2009, p. 86). However, it did not present the author's comprehensive systematic concept yet. Rather, Florensky's contribution was to help the listeners challenge the traditional scholastic-rationalist vision of the world, which was completely foreign to him. Namely, according to the author the inability to find an adequate concept for something does not confirm the nonexistence of such a being, on the contrary, it is a reference to its greatness, mystery, and sanctity (Florensky 2000, vol. 3(2), p. 148–49). The abovementioned subject of the mystery and sublime being the author managed to develop in a more systematic way in his work *Philosophy of Cult*, and met his goal in the article *Cult, Religion and Culture* which we examined in more detail. The mystery and sanctity of being helps us in the historical process to constantly and again rediscover the very cult of God, which is, and will always be an invaluable treasury of humanity (Hospodár 2010, p. 23).

At the beginning of the 20th century, sacral theory made a clever attempt to replace this treasury of humanity with a knowledge of the cult but without the participation in it. Although sacral theory contradicts religion in its essence, in its statement, according to Florensky, it rightly points to the unavoidable need for a cult in the life of a man who is not only a thinking individual or a formless number, but a man, a person whose integrity is achieved thanks to unity with everyone in the cult. Just like most Russian religious philosophers, Florensky also once again points to the ontological unity with all, the so-called *sobornost'*, the choral beginning or synergism, which is deeply rooted in man precisely because of the cult. "*A person here means everything, but always in all and in the unity of all, and outside that 'all' the person as such is nothing.*" (Florensky 2004, p. 77)

There is no coincidence when he calls man *homo liturgus*, because from the cult as the primordial activity of man, the other activities are derived. They depend on the real cult. What happens when the true cult of God–Man Jesus Christ is rejected? Undoubtedly, there will be consequences in an individual and social life and in the historical process. Florensky's *Philosophy of Cult* thus presents the cult in a special unity with the philosophy of history, a unity which, with its interesting elaboration and an application to the present era, offers an inspiration for the constant search for the common ground between this fleeting world and the eternal world. Man's behavior often resembles the cult, even when he deviates from the true cult, and history confirms that when he leans toward materialism, he creates a cult of matter, if, on the other hand, he prioritizes reason, and he creates a cult of ideologism. This observation also appears with contemporary authors from various disciplines, unbelievably at such field as marketing, where the terms cult and religion are applied: Marketing topics shape human thinking and actions to worship the "cult of the moment", and socialize individuals in favor of behavioral strategies based on "instant gratification" (Roubal 2014, p. 111).

Florensky convinces us, both with his life and with his work, that there is an undeniable 'kinship' relationship between history and eternity, and the cult is its unmistakable testimony lifting the timeliness to the eternity. History has its primordial image or prototype in the cult of the living God, i.e., in the real revelation of God in space and time. Without the revealing God in the cult, history would be both unthinkable (*notiones*) and unrealizable (*instrumenta*) for the benefit of man, the real and not chimerical benefit to man who is created in God's image and likeness.

**Funding:** This research received no external funding.

**Conflicts of Interest:** The author declares no conflict of interest.

## Notes

[1] (διὸ καλῶς ἀπεφήναντο τἀγαθόν, οὗ πάντ᾽ ἐφίεται. διαφορὰ δέ τις φαίνεται τῶν τελῶν: τὰ μὲν γάρ εἰσιν ἐνέργειαι, τὰ δὲ παρ᾽ αὐτὰς ἔργα τινά. ὧν δ᾽ εἰσὶ τέλη τινὰ παρὰ τὰς πράξεις, ἐν τούτοις βελτίω πέφυκε τῶν ἐνεργειῶν τὰ ἔργα).

[2] (τὸ γὰρ ἔργον τέλος, ἡ δὲ ἐνέργεια τὸ ἔργον, διὸ καὶ τοὔνομα ἐνέργεια λέγεται κατὰ τὸ ἔργον καὶ συντείνει πρὸς τὴν ἐντελέχειαν).

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
