# Peer review of "From the Philosophy of Cult to the Philosophy of History in the Work of Pavel Alexandrovich Florensky (* 1882 + 1937)"

_religions, doi:10.3390/rel12070533_

Round 1
Reviewer 1 Report
- The argument is not clearly presented, giving the impression there are too many preambles before coming to the point of the article.
- What is the originality of the text wand what does it contribute to the Florenskian studies.
- Surely the article should be re-written in better English idiom. The article gives the impression that it is actually a translation from one or possibly more Italian published texts. Hence the high suspicion of plagiarism direct or indirect.
- There are typos to be corrected.
Author Response
Thank you for your opinions and suggestions to help improve my manuscript. I would like to respond and address the comments in the reviews on my work From the Philosophy of Cult to the Philosophy of History in the work of Pavel Alexandrovich Florensky.
First, I would like to emphasize that I wrote this article using the primary sources in Russian language and it is not a plagiarism or a translation of an Italian text. To say that it is “full of Italianisms” is, to me, both surprisinging and inexplicable. Only in the introduction to the article I used the Italian translation as it is shown by the following quotes: Piovesana 1992; Valentini 2000, Valentini 2004, Šentalinsky 1994.
As for the statement on the originality of the content of the text, it is a matter of the opinion of each of the reviewers to declare something original or not in accordance with their knowledge and conscience. However, I defend the originality and consider necessary to give the following proofs:
You mentioned several publications in Italian. I am familiar with all of them and neither of these contains the work Cult, Religion and Culture or comments on it. In Tagliagambe 2006, chapter 8 , pp. 99-108 there is no mention of it. The same we can say about the anthology “Il Pensiero Polifonico di Pavel Florenskij” (2018). Also in Valentini's Bellezza e Liturgia (2010) we find five of Florensky's writings from years 1909-1923, namely: L’ortodossia, Il rito come sintesi delle arti, Nota sull’ortodossia, Cristianesimo e cultura, Lezioni sulla concezione cristiana del mondo. Again, the work Kult, relígia i kultúra I wrote on, was not published nor commented there. Further, when talking about Valentini's Il Significato dell'Idealismo (2012); Russian title Smysl idealizma (1914),(Lezioni propedeutiche a una serie di conferenze dedicate alla storia del platonismo, tenute agli studenti del primo anno dell’Accademia Teologica Moscovita (ATM)), even in this publication, there is no mention of the material in question. Finaly, Florensky's work Lo spazio e il tempo nell'arte (1993; 1995) was written during years 1924-25. My manuscript deals with the original of Kult, relígia i kultúra first written on the 7th of November 1917 and later revised on the 5th of May 1918 and is not included in the work from 1924-25 nor is found in the Italian translations mentioned above.
Further, the structure for the text together with a short description of each chapter is included in the Introduction part. The structure I chose is based on the argument that the cult is the primary activity of man and it is only the cult of the True God which shapes man and gives his culture the right direction. I divided the argument into four (not counting the Introduction and Conclusion) major parts and gave them each a title to show their main goal. I want to point out that Florensky in his original text Kult religia i kultura used simple paragraphs and just numbered them from 1 to 27. I think that the text structure chosen for my article serves the aim, marks the process of argumentation and introduced the topic to the reader at the beginning and in the last paragraphs of the Conclusion.
The major contribution of the article to the Florenskian studies is the presented intersection of the philosophy of the cult and the philosophy of history. Florensky addresses the work to his contemporaries in turbulent times and understands what lies at the very core of human existence and the historical development. He writes that the cult is the primary activity of man and at the same time a gift offered to him by God. He made it very clear that the cult as a gift has to be accepted. More than a hundred years later we see that man lives in the culture of pseudocults, his own creations, and labels them as progress. It is no coincidence that Florensky begins his work with metaphysics. The cult is at the basis of human existence, and man needs to recognize the divine-human balance within himself and act according to it. The history will always reflect either this balance or its opposite.
Concerning the language comments, I made the corrections with the help of a native English speaker and removed the Greek quotes from the main body of the text and provided their official English translation. To my best knowledge, the English version of Kult religia i kultura was not published. If there were such translation, I would see a great opportunity to compare it with the original I am working with. As I have already mentioned in my manuscript reading and interpreting Florensky is a demanding activity for his unique style and language. I am convinced that my attempt to present his original work to the English speaking readers is worth your attention.
Reviewer 2 Report
The text needs some stylistic correction: are some strange formulation, like "Philosopher Plato" instead of just "Plato" or "philosophers Sophocles..."; some unnecessary repetitions ("At first, Father Pavel vehemently opposed the false accusation against him, but after realizing that his confession would release some of his fellow prisoners from prison, he accepted the false accusation.")
The author should cite Aristotle's "Metaphysics" according to English translation. Also Florensky's work ("The Pillar and Ground of the Truth") and the book by Lossky, Zenkovsky, Men and others have been published in English - the author has to consider to cite them basing on English translations. There is no need to double Florensky's work (the first and the subsequent editions).
Author Response
I made the corrections with the help of a native English speaker and removed the Greek quotes from the main body of the text and provided their official English translation. To my best knowledge, the English version of Kult religia i kultura was not published. If there were such translation, I would see a great opportunity to compare it with the original I am working with. As I have already mentioned in my manuscript reading and interpreting Florensky is a demanding activity for his unique style and language. I am convinced that my attempt to present his original work to the English speaking readers is worth your attention. It is, however, the message of the work, not the linguistic perfection, I am aiming at.
I am ready to improve my work and I am open for the constructive dialogue. If there are any further suggestions, requests for explanation of certain parts of the text or questions I am eager to answer them.
Round 2
Reviewer 1 Report
English language and style must be improved.
I suggest also referencing the following studies to show in what way the present artcile is a step forward and contribution on what has been already stated in other studies.
What about the idea of proletkult in such a discussion, considering Florenskij's context and the fact the author starts with a reference to the Bolshevic Revolution?
It seems to me that the present notions in the article are a reworking of an already published article and not so much a step further as indicated in the article itself. Also the Introduction is too long. I wouldn't advise also self-quotations from: Porubec, Daniel. 2019. Filozofia kultúry z perspektívy kultu v diele Pavla Alexandroviča Florenského (*1882,+1937). Studia Theologica 704/2: 197–224.
Florensky discusses also the human person from the perspective of homo faber in this important work which is not cited or mentioned in this article: "U vodorazdelov mysli. Čerty konkretnoj metafiziki".
I suggest to read Prof. Sandro Lanfranco' s contribution on the uniqueness of man in connection with Florensky's reflection on man as a tool-using creature: "Our brain and its mind have not only created tool use, they have also created our gods. And it is this particular product of our cognition that might actually be invoked for our uniqueness." in 'Pavel Florensky and the Uniqueness of Man', in Melita Theologica 69/1 (2019) 23-34.
I would suggest also to have a look at:
Boneckaja N. K. 2010. «Homo faber» i «Homo liturgus» (Filosofskaja antropologija o. P. Florenskogo) in: Voprosy filosofii, vol. 3, pp. 90–109.
Glen Attard, Closest to the Heart. A Mystagogy of Spiritual Friendship in Pavel A. Florenskij's The Pillar and Ground of Truth, Malta: Horizons 2020. 651pgs. This study deals with the theme of cult. Also this study was judged by Robert Slesinski as "a defining one in present-day Florenskij scholarship"
In line 524, there is a great generalisation regarding poets and commentaries. What about the great mystics like John of the Cross who infact wrote commentaries to his lyrical poetry, both of great linguistic and theological value? Is your argument held by Florensky?
Author Response
Dear reviewer,
Thank you for your suggestions.
The English language in my manuscript has already been checked and corrected by a native English speaker.
The contribution of the article is in the already mentioned original connection between the philosophy of the cult and the philosophy of history. Florensky associated the philosophical and theological concepts with the philosophy of history in a unique way and explained the correlation in the three given examples. The astonishing similarity to the modern times, where people create various kinds of cults (e.g. the mentioned cult of the moment), makes his work universal and valid. Personally, the main inspiration and reason for writing my article was Florensky's work and his approach of the cult and history, and I only marginally touch on current pseudocults.
Florensky was the most apolitical author of the Russian religious philosophers. The idea of proletarian cult, political religions or Marxism was elaborated in detail by other authors such as N. A. Berdyaev, B. P. Vysheslavcev and S. N. Bulgakov. However, it is not excluded that Florensky could write his Kuľt, religia i kultura (1917-1918) in the foresighted anticipation of the coming Bolshevik pseudocult.
The self-quotation from Studia Theologica 704/2: 197–224, Porubec, Daniel. 2019. Filozofia kultúry z perspektívy kultu v diele Pavla Alexandroviča Florenského (*1882,+1937) was suggested by the editor and had been previously consulted.
I have quoted from "U vodorazdelov mysli. Čerty konkretnoj metafiziki" (1922) at three different places in my article, see the reference Florensky, Pavel Alexandrovich.2000. Sočinenija. vol. 3(2). Moskva: Mysľ, p.3,13. I have added two other quotes from the source on pages 4 and 12.
In his short article Homo faber from the mentioned "U vodorazdelov mysli. Čerty konkretnoj metafiziki" (1922), and Florensky, Pavel Alexandrovich.2000. Sočinenija. vol. 3(1). Moskva: Mysľ. pp. 374-382., Florensky analyses the antinomic relationship between nature and culture. He explicitly solves the problem of creating culture by using the tool of human knowledge, namely, the terms. The issue of the cult, however, does not even once occur here.
Boneckaja in her article «Homo faber» i «Homo liturgus» criticizes Florensky and supports her assertion with Berdyaev's statements about Florensky's anthropology where the Kantian "order of nature" harmfully rules over the "order of freedom", that is, nature rules over culture. Florensky, on the other hand, talks about their mutual cooperation (performance in unison) and does not prioritize either of them.
It seems to me that in the article 'Pavel Florensky and the Uniqueness of Man' Prof. Sandro Lanfranco in the last two sentences of the conclusion also inclines toward the "order of nature". Certainly, I could not study deeply all of the work in the given timeframe.
We can find more on the issue of the cult and Homof aber from the Florensky's perspective in other articles as well. If there is, in your opinion, any relevant need for citing from other authors I am open to the option.
The corrections made in the text:
- 3 - (Glen Attard 2020, p. 219–222)
- 3 - (Glen Attard 2020, p. 214–218)
- 4 –( Florensky 2000, Homofaber, vol. 3(1) p. 378-379)
- 4 - (Lanfranco 2019, p. 23–34; Boneckaja 2010, p. 90-109)
- 12 – (Florensky 2000, U vodorazdelovmysli, vol. 3(2), p. 381–383, 439–441).
- 14 – References.
Sincerely,
Daniel Porubec
Round 3
Reviewer 1 Report
I went through the revision made by the author of the article. It has been improved and I think it is publishable.
The issue of plagiarism suspicion might be also linked to the way the author expressed himself in English and the way some ideas were presented in connection with his other previous article already published in another journal.
I think now the author has proved enough that he is the original author of the contribution.
Hence I think that the article should make it to publication.
Kindly extend my best wishes to the author.